# Modified Hydrothermal Route for Synthesis of Photoactive Anatase TiO₂/g-CN Nanotubes from Sludge Generated TiO₂

**Sayed Mukit Hossain** [1], **Heeju Park** [2], **Hui-Ju Kang** [3], **Jong Seok Mun** [3], **Leonard Tijing** [1], **Inkyu Rhee** [4], **Jong-Ho Kim** [2], **Young-Si Jun** [2,*] **and Ho Kyong Shon** [1,*]

1 Centre for Technology in Water and Wastewater, School of Civil and Environmental Engineering, University of Technology, P.O. Box 123, Broadway, Sydney NSW 2007, Australia; sayed.m.hossain@student.uts.edu.au (S.M.H.); leonard.tijing@uts.edu.au (L.T.)

2 School of Chemical Engineering, Chonnam National University, Gwangju 61186, Korea; point1014@hanmail.net (H.P.); jonghkim@chonnam.ac.kr (J.-H.K.)

3 Department of Advanced Chemicals & Engineering, Chonnam National University, 77 Yongbong-ro, Buck-gu, Gwangju 61186, Korea; gmlwn120@gmail.com (H.-J.K.); answhdtjr8726@gmail.com (J.S.M.)

4 Department of Civil Engineering, Chonnam National University, Gwangju 61186, Korea; rheei@chonnam.ac.kr

* Correspondence: youngsi.jun@gmail.com (Y.-S.J.); Hokyong.Shon-1@uts.edu.au (H.K.S.); Tel.: +61-447-332-707 (H.K.S.)

**Abstract:** Titania nanotube was prepared from sludge generated TiO₂ (S-TNT) through a modified hydrothermal route and successfully composited with graphitic carbon nitride (g-CN) through a simple calcination step. Advanced characterization techniques such as X-ray diffraction, scanning and transmission electron microscopy, infrared spectroscopy, X-ray photoelectron spectroscopy, UV/visible diffuse reflectance spectroscopy, and photoluminescence analysis were utilized to characterize the prepared samples. A significant improvement in morphological and optical bandgap was observed. The effective surface area of the prepared composite increased threefold compared with sludge generated TiO₂. The optical bandgap was narrowed to 3.00 eV from 3.18 in the pristine sludge generated TiO₂ nanotubes. The extent of photoactivity of the prepared composites was investigated through photooxidation of NOₓ in a continuous flow reactor. Because of extended light absorption of the as-prepared composite, under visible light, 19.62% of NO removal was observed. On the other hand, under UV irradiation, owing to bandgap narrowing, although the light absorption was compromised, the impact on photoactivity was compensated by the increased effective surface area of 153.61 m²/g. Hence, under UV irradiance, the maximum NO removal was attained as 32.44% after 1 h of light irradiation. The proposed facile method in this study for the heterojunction of S-TNT and g-CN could significantly contribute to resource recovery from water treatment plants and photocatalytic atmospheric pollutant removal.

**Keywords:** NOₓ removal; TiO₂ nano tube; nitrate selectivity; TiO₂/g-CN; optical bandgap modification

## 1. Introduction

Over the last few years, wastewater generation has increased at an exponential rate. The increase in the global population, which makes the biggest contribution to the depletion of wastewater, remains a major issue for water security [1]. While the adoption of wastewater recycling methods has successfully addressed many of the pollution problems that result from the introduction of polluted water into the natural environment, the hazardous sludge that is generated during the physiochemical

treatment of wastewater remains a major concern [2,3]. In recent times, Shon, et al. [4] described a cutting-edge method of sludge reuse that involved the synthesis of titania from Ti salt-based flocculated sludge. Through the use of Ti-salts, viz., titanium tetrachloride ($TiCl_4$), titanium sulfate ($Ti(SO_4)_2$), and polymerized $TiCl_4$/ $Ti(SO_4)_2$, within water flocculation, it is possible to calcine the sludge to generate $TiO_2$, which represents a useful byproduct [5–7]. Previous studies found that approximately 40 mg of anatase titania can readily be produced from the flocculation and calcination of a liter of secondary sewage effluent [8]. As such, it was concluded that Ti-based salts as coagulants represent an effective means of addressing some of the environmental concerns that are linked to the generation of large volumes of sludge from conventional coagulants [3,9,10].

Researchers have concentrated their attention on titanium dioxide ($TiO_2$) because it provides a greater degree of stability, high photoactivity, wide surface area, commercial availability, and non-toxicity [11,12]. Moreover, $TiO_2$ has been utilized in a range of applications including solar cells [13], paints [14], photocatalysis [15], and UV shielding [16,17]. Within these various contexts, $TiO_2$ is deemed to represent one of the most attractive semiconductors in the field of photocatalysis because of its ability to limit environmental pollution [17]. A broad range of $TiO_2$ morphologies has been explored in the current literature to improve the use of photocatalysts as suspension or photoelectrodes. Subsequently, one of the most frequently explored morphologies is aligned $TiO_2$ nanotubes (TNTs) [18–20]. The significant focus on TNTs could be attributed to their improved reactivity and desirable structural geometry. Specifically, the photoconversion productivities of nanotubular arrays of $TiO_2$ have been found to be particularly high because of the orthogonal carrier separation and the high charge transfer rate [19]. A simple TNT synthesis process was proposed by Kasuga, et al. [21] through the hydrothermal treatment of titania nanoparticles in an NaOH-based aqueous solution and reported the formation of nanotubes with an effective surface area of 400 $m^2$/g having a diameter of 8 nm.

As TNTs have a wide variety of applications, they were expected to have a more beneficial impact on photocatalytic applications than other forms of titania such as nanoparticles, colloidal, and alternative structures. [22,23]. Some researchers have hypothesized that the as-synthesized Na/H-titanate nanotubes can degrade organic pollutants at a higher level of efficiency than alternative methods [24,25]. The augmentation in photocatalytic activities of the TNTs are commonly attributed to their tubular structure, increased rate of sedimentation, and augmented effective surface area [23,25]. Additionally, Okour, et al. [25] investigated the post-treatment of titanate nanotubes as a means of preparing thiourea-doped titanate nanofibers. They found that thiourea-doped TNTs delivered superior performance in decaying acetone under UV and visible light. They attributed this performance to the fact that nanotubes have a smaller crystallite and larger specific surface area.

Nevertheless, $TiO_2$ can only be excited by UV light at wavelengths below 380 nm because it exhibits a wide bandgap [16]. Unfortunately, only around 3–5% of sunlight contains UV light under 400 nm; as such, there is a low efficiency of solar energy conversion [26,27]. Hence, researchers have employed non-metal dopants, for example, nitrogen (N) [28,29] and sulfur (S) [16], to increase the photoactivity of $TiO_2$ within the visible light region. Studies revealed that N-doped atoms reduce the optical bandgap of $TiO_2$, and thereby enhance its responsiveness to visible light irradiation [30]. There is a wide belief that N atoms act as a viable replacement for lattice oxygen atoms in a $TiO_2$ structure [16,31]. They serve to modify the bandgap of the catalyst such that it is within the visible light region. There is a potential to combine the impact of multiple dopants where dopants are renowned for changing the structure of $TiO_2$, as doing so can enhance photooxidation activity under visible light [29]. In addition, heterojunctions can be constructed by merging $TiO_2$ with alternative semiconductors that exhibit a narrow bandgap to explore an alternative pathway by which it is possible to increase the visible light response of $TiO_2$-based catalysts [27,32]. Heterojunctions are capable of producing an effective photogenerated charge separation between the $TiO_2$ substrate and the associated semiconductors in a similar manner to dye sensitized $TiO_2$. This plays a vital role in improving the performance of photocatalysts based on semiconductors.

Lately, as a heterojunction constituent of $TiO_2$, the 2D graphitic carbon nitride (g-CN) has been found to attract growing attention owing to its suitable optical band edge positions, superior chemical stability, and the light absorption capacity in the visible light spectrum [15,33,34]. When composite photocatalyst is prepared using anatase $TiO_2$ and 2D g-CN, the optical band edge positions (g-CN: −1.1 eV and +1.6 eV; $TiO_2$: −0.1 eV and +3.1 eV) of the constituents remain intact, and a type II heterojunction interface is formed [35,36]. Therefore, during light irradiation, it is possible for the photoinduced electrons ($e^-$) and holes ($h^+$) to travel between the valency band (VB) and conduction band (CB) of $TiO_2$ and g-CN [27]. This serves to result in extended charge carrier separation within the heterojunction. Multiple studies have attempted to synthesize $TiO_2$/g-CN heterostructures as a means of achieving a wide light absorption range and high photoinduced charge separation productivity through the use of a heterojunction approach [27,35,37].

One of the most significant environmental issues of the contemporary era concerns the atmospheric pollution that is caused by nitrogen oxides ($NO_x$). It has been found that anthropogenic activities, such as burning fossil fuels and the denitrification of nitrate salts from the topsoil, are the primary atmospheric $NO_x$ sources [32]. Hence, there has been a notable increase in the concentration of atmospheric $NO_x$ over the last few decades. This is evident in the dense haze that can be observed because of the use of secondary aerosols. Thankfully, compared with conventional atmospheric $NO_x$ removal techniques, photocatalysis can readily remove gaseous $NO_x$ at a significantly lower concentration (ppb level), which mimics the ambient atmospheric condition [34,35]. Moreover, the removal of atmospheric $NO_x$ through photooxidation could facilitate sustainability by using the large-scale application of renewable solar energy at mild reaction conditions [34].

To date, a number of studies have been conducted to prepare the heterojunction between titanate nanotubes and g-CN using commercially available precursors for $TiO_2$ sources, such as tetrabutyl titanate [18,22], Ti foils [19], and commercial $TiO_2$ powders [38]. Insufficient amounts of studies have been found concerning the synthesis of single/composite photocatalysts by using the sludge generated $TiO_2$ [3,10,39]. To the best of the author's knowledge, this is the first study to achieve an effective heterojunction of titanate nanotubes formed by sludge with g-CN. Consequently, the objectives of the study were to evaluate the feasibility of fabricating heterojunction between sludge generated $TiO_2$ nanotubes and melamine exfoliated g-CN for superior atmospheric NO removal under an extended irradiation spectrum. Initially, for removing impurities from sludge generated $TiO_2$ and the fabrication of H-titania nanotubes (H-TNTs), a modified hydrothermal approach was employed. Later, to enhance visible light absorption of the as-prepared S-TNTs, a one-step template-free route was utilized to facilitate heterojunction between 1D S-TNTs and 2D g-CN. To assess the morphological and optical attributes of the as-prepared samples, advanced characterizations were carried out. Besides, following ISO 22197-1 (2007) [40] and ISO 17198-1 (2018) [41], the $NO_x$ photooxidation potentials of the as-prepared samples were evaluated under UV and visible light, respectively.

## 2. Results and Discussion

### 2.1. Morphological Attributes

X-Ray Powder Diffraction (XRD) and Brunauer–Emmett–Teller (BET) Results

To understand the crystallographic structure and phase, X-ray powder diffraction (XRD) patterns were obtained for the as-prepared samples and compared with standard charts. Figure 1a compares the XRD patterns of the prepared composites S-TNT1 and S-TNT2 with prepared H-TNT, S-TNT, S-$TiO_2$, and g-CN. As shown in Figure 1a, the prepared S-$TiO_2$ was perfectly indexed as anatase titania (JCPDS No. 21-1272) [18,38]. The XRD pattern of the prepared nanotubes (S-TNT, STNT1, and STN2) shows that the phase structure and the crystallinity of S-$TiO_2$ remained intact after the modified hydrothermal treatment [25]. However, before calcining H-TNT at 550 °C into S-TNT, the crystallographic development of anatase titania was insignificant. Scherrer's equation was utilized to estimate the crystal size of the anatase $TiO_2$ at the crystal plane (101) [18]. The calculated crystal sizes

were 15.40 nm, 16.44 nm, 15.79 nm, and 15.83 nm for S-TiO$_2$, S-TNT, S-TNT1, and S-TNT2, respectively. Hence, nanotube formation and composite preparation by utilizing S-TiO$_2$ showed negligible effects on the crystal size and phase. One interesting finding was the small peak around the 2θ value of 27.5°, which denotes the formation of little amounts of rutile (110) phase in S-TiO$_2$. According to the study conducted by Liao, et al. [42], anatase titania started to be converted to the rutile phase at the temperature above 550 °C, which is in coherence with the formation of the rutile phase in S-TiO$_2$. However, because of the presence of various impurities in the Ti incorporated sludge, the phase transformation is negligible even at 600 °C. Besides, Shon, et al. [4] argued that, while calcining Ti induced sludges, rutile titania becomes the dominant form of titania when the temperature is above 800 °C. Nevertheless, calcining with a melamine level of impurities increased in the precursor may have diminished the peak around 27.5° in the as-prepared samples (S-TNT, S-TNT1, and S-TNT2).

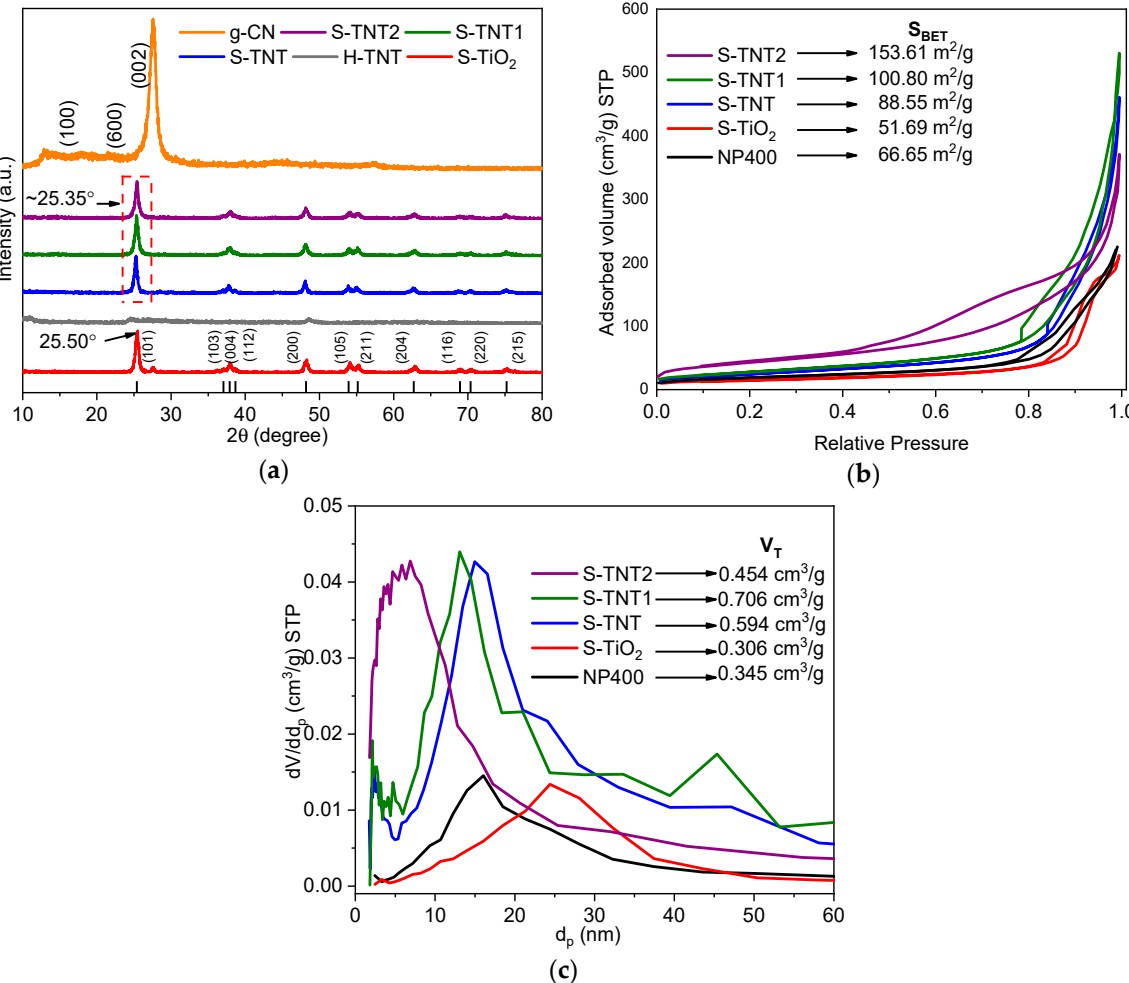

**Figure 1.** (**a**) X-ray powder diffraction (XRD) patterns of the as-prepared samples S-TiO2, H-TNT, S-TNT, S-TNT1, and S-TNT2; N$_2$ adsorption-desorption isotherms (**b**) and the pore size distribution (Barrett–Joyner–Halenda (BJH) adsorption) curves (**c**) of commercially available NP400 along with as-prepared sludge generated titanium dioxide (S-TiO$_2$), sludge generated TiO$_2$ nanotube (S-TNT), S-TNT1, and S-TNT2.

In comparison, the prepared g-CN nanosheets showed characteristics peaks around 2θ of 13.10° and 27.50°, which represent the Miller indices (hkl) value of (100) and (002), respectively [43]. The crystal planes, as mentioned earlier, correspond to the periodic in-plane arrangement of N-bridged tri-*s*-triazine repeating units (100) and the graphitic stacking of the conjugated aromatic system (002), with an estimated lattice spacing of 0.693 nm and 0.324 nm, respectively [44]. Because of the negligible

loading of g-CN, no visible characteristic peaks were observed in the prepared composites S-TNT1 and S-TNT2 [45]. However, the composite formation of S-TNT with g-CN showed a negative shift in 2θ from 25.50° to 25.35° for the anatase (101) plane, which may have indicated the successful formation of S-TNT/g-CN composite [26].

Brunauer–Emmett–Teller (BET) analysis was conducted with the help of $N_2$ adsorption-desorption isotherm to assess the amendments in porosity due to nanotube and nanotube-based composite formation by employing $S-TiO_2$. For the as-prepared samples, graphs for the adsorbed volume of $N_2$ (@STP) versus relative pressure ($P/P_0$) are demonstrated in Figure 1b. Additionally, for better comparison, BET assessment results of a commercially available anatase $TiO_2$ (NP400) were compared with the prepared $S-TiO_2$. Along with NP400, all the prepared samples showed type IV(a) isotherm, according to International Union of Pure and Applied Chemistry (IUPAC) categorization, which confirms the existence of mesopores in the samples [46]. Additionally, the prepared $S-TiO_2$ showed a type H2 (b) hysteresis loop, wherein cavitation-oriented evaporation in slender array of pore necks could be the possible reason. However, after the formation of nanotubes (S-TNT), the hysteresis loops shifted to type H3, and later, the incorporation of g-CN with S-TNT changed the hysteresis loop to type H4 [46]. Moreover, it seems that an increase in the BET surface area of S-TNT2 was due to micropores' development compared with the other samples. Hence, the composite S-TNT2 was presumed to be a combination of micro and mesoporous structure [46], which is further clarified by the pore size distribution curves depicted in Figure 1c.

Using the adsorption data and the Barrett–Joyner–Halenda (BJH) method, the pore size distributions of the corresponding samples are depicted in Figure 1c. The effective BET surface area ($S_{BET}$) of $S-TiO_2$ and NP400 was determined to be 51.69 $m^2/g$ and 66.65 $m^2/g$, respectively. A slight reduction in the $S_{BET}$ value of $S-TiO_2$ can be explained from the extended pore size distribution over a larger pore diameter (>20 nm). On the other hand, after the fabrication of S-TNT from $S-TiO_2$, the $S_{BET}$ values increased to 88.55 $m^2/g$, and the pore size distribution shifted towards a lower pore diameter, with a maximum pore diameter of 15 nm. Later incorporation of g-CN further increased the surface areas to 100.80 $m^2/g$ and 153.61 $m^2/g$ for S-TNT1 and S-TNT2, respectively (see Figure 1b). Moreover, the larger effective surface area of S-TNT2 supports the shifting of pore size distribution curve towards lower pore diameter, and the maximum pore diameter was found to be half (5.82 nm) compared with S-TNT1 (12.90 nm).

## 2.2. Electron Microscopy (SEM and TEM Images)

Figure 2 and Figure S1 show the scanning electron microscope (SEM) images and energy-dispersive X-ray (EDX) spectra of the as-prepared samples, respectively. The SEM image of the prepared $S-TiO_2$ (Figure 2a) showed somewhat dispersed spherical-shaped particles and the EDX spectra confirm the existence of $TiO_2$ along with the presence of a considerable amount of C (9.56 at.%) atoms (Figure S1a), which is in coherence with previous studies, where the source of these C has been presumed as the organic contents of the precursor wastewater [3,4]. After the hydrothermal treatment, the prepared H-TNT also showed a high carbon content along with the formation of nanotubes with a length of several μm and a diameter of 50 to 100 nm (Figure 2b). Further calcination at 550 °C for 3 h seems to preserve the nanotubes' features and showed enhancement in tube length of the prepared S-TNT (Figure 2c).

Moreover, extended calcination seems to reduce the content of C (3.29 at.%) in S-TNT, which may have significant impacts in photoactivity of the prepared S-TNT (see Figure S1c). The SEM images of the as-prepared S-TNT1 and S-TNT2 show that the nanotube features of the H-TNT precursor remained intact with further length extension and surface smoothening of the nanotubes. Nevertheless, from the SEM image, it is hard to indicate the presence of 2D g-CN plates in the prepared composites (S-TNT1 and S-TNT2). Presumably, some of the 2D g-CN nanosheets were present covering the surface of the prepared nanotubes [18]. Moreover, the EDX analysis of S-TNT1 (Figure S1d) and S-TNT2

(Figure S1e) confirmed the presence of small amounts of C (19~40 at.%) and N (2~5 at.%) atoms in the prepared composites.

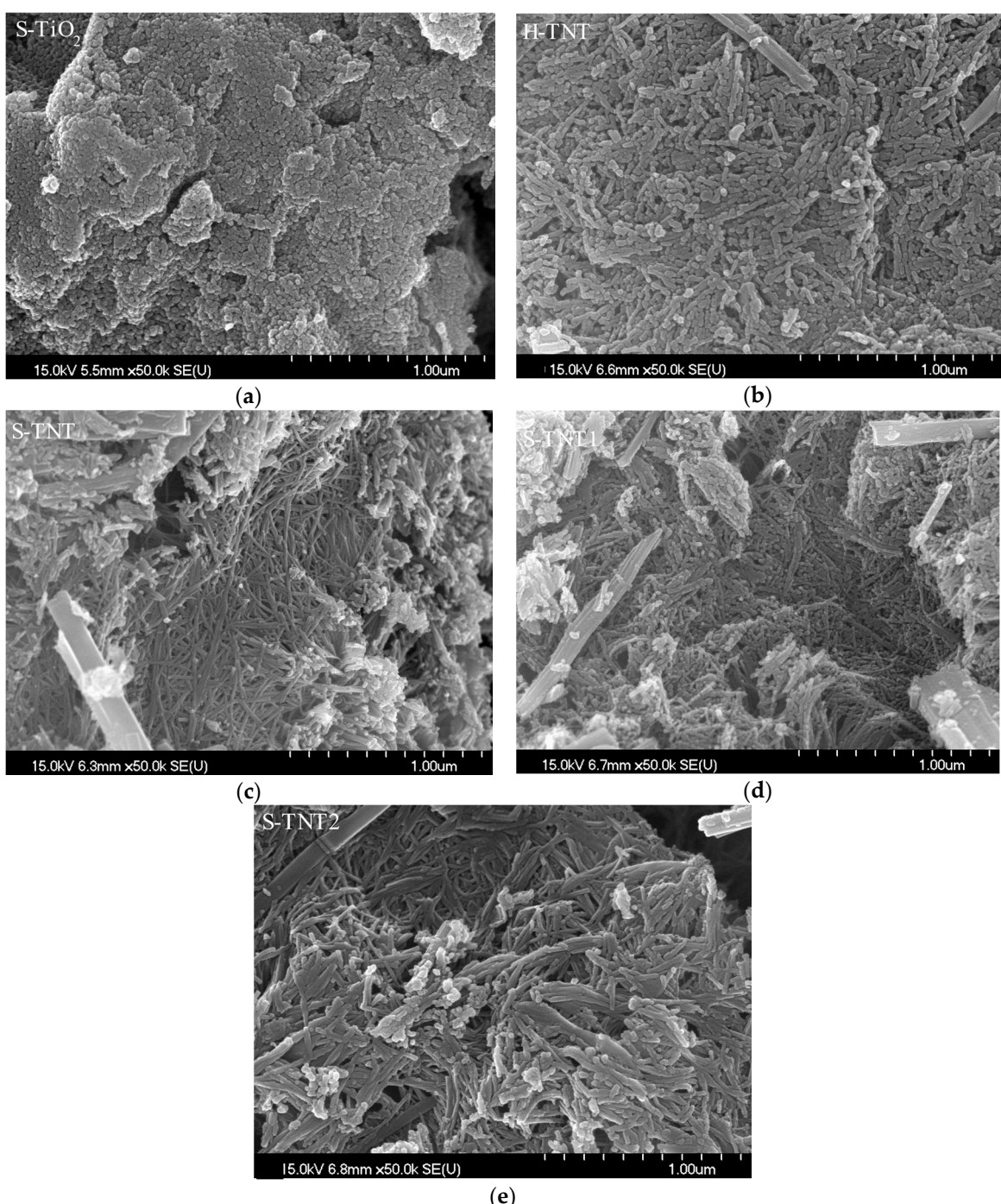

**Figure 2.** Scanning electron microscope (SEM) images of (**a**) S-TiO$_2$, (**b**) H-TNT, (**c**) S-TNT, (**d**) S-TNT1, and (**e**) S-TNT2.

To further assess the microstructure of the as-prepared S-TiO$_2$ and confirm the presence of g-CN in the prepared composite (S-TNT2), transmission electron microscopy (TEM) analysis was conducted, and the corresponding images are illustrated in Figure 3. The average crystal size of the as-synthesized S-TiO$_2$ depicted in Figure 3a corresponds well with the aforementioned XRD spectra of S-TiO$_2$. Moreover, as shown in Figure 3b, lattice fringes were found with the distances of 0.352 nm, 0.238 nm, and 0.188 nm, which confirm the presence of anatase (101), (004), and (200) planes,

respectively [22,45]. From the TEM image of S-TNT (Figure 3c), the presence of g-CN nanosheets can easily be found in/on the surface of S-TNT. Similar, lattice fringes as S-TiO$_2$ were found in the TEM image of S-TNT2 (Figure 3d); additionally, lattice fringes with the distance of 0.318 nm were found, confirming the presence of g-CN (002) nanosheets [43,44].

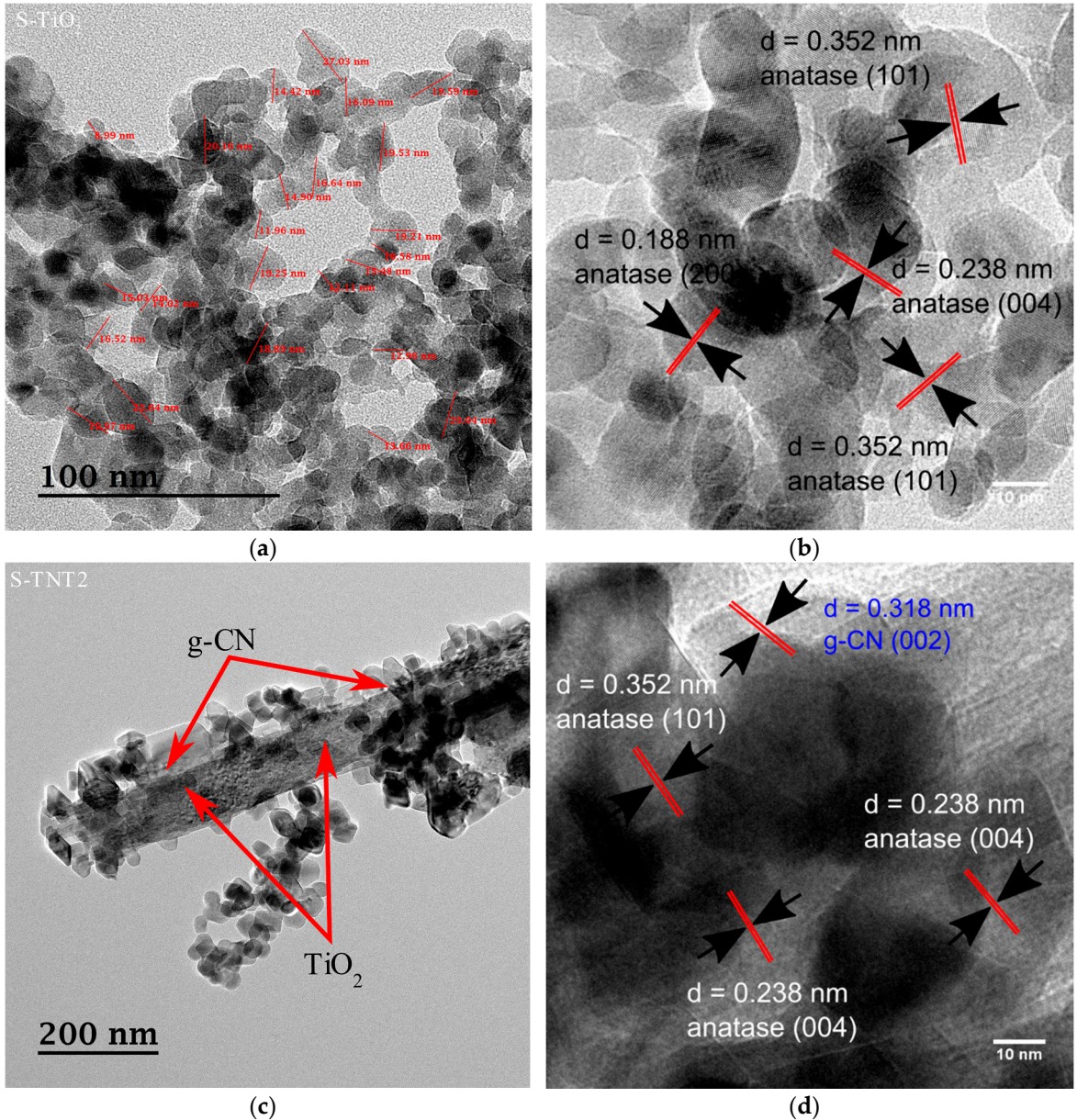

**Figure 3.** Transmission electron microscopy (TEM) images of the as-prepared S-TiO$_2$ (**a,b**) and S-TNT2 (**c,d**)

## 2.3. Fourier Transform Infrared (FT-IR) and X-Ray Photoelectron Spectroscopy (XPS) Spectra

FT-IR spectra of the as-prepared composites and the single-component samples were compared to investigate the chemical structure for the validation of the presence of TiO$_2$ and g-CN. As shown in Figure 4a, except for g-CN, all the samples showed a broad absorption peak around 450–600 cm$^{-1}$, which can be assigned to Ti-O-Ti stretching modes and confirms the presence of TiO$_2$ in the samples [47,48]. Another broad absorption peak was observed within the range of 3000–3650 cm$^{-1}$, which could be due to the stretching mode of O-H generated from the adsorbed water on the samples [48]. For pure g-CN, the peaks around 809.70 cm$^{-1}$ and 3200 cm$^{-1}$ could be dedicated to the stretching mode of *s*-triazine rings and N-H stretching of the remaining NH$_x$ (x = 1, 2) groups [44]. As shown in

Figure 4a, a series of absorption bands were found within the wavelength range of 1200–1600 cm$^{-1}$, which are due to the N-containing carbon of the core aromatic rings of g-CN [18]. A zoomed-in image was provided within Figure 4a to identify the increasing N content via the addition of a greater g-CN content in synthesis processes. Although, through EDX analysis, we have confirmed the presence of a small amount of C and N atoms in S-TNT1, it showed no characteristics peaks for g-CN. On the other hand, S-TNT2 showed the apparent presence of g-CN and confirmed the successful preparation of the S-TNT/g-CN composite.

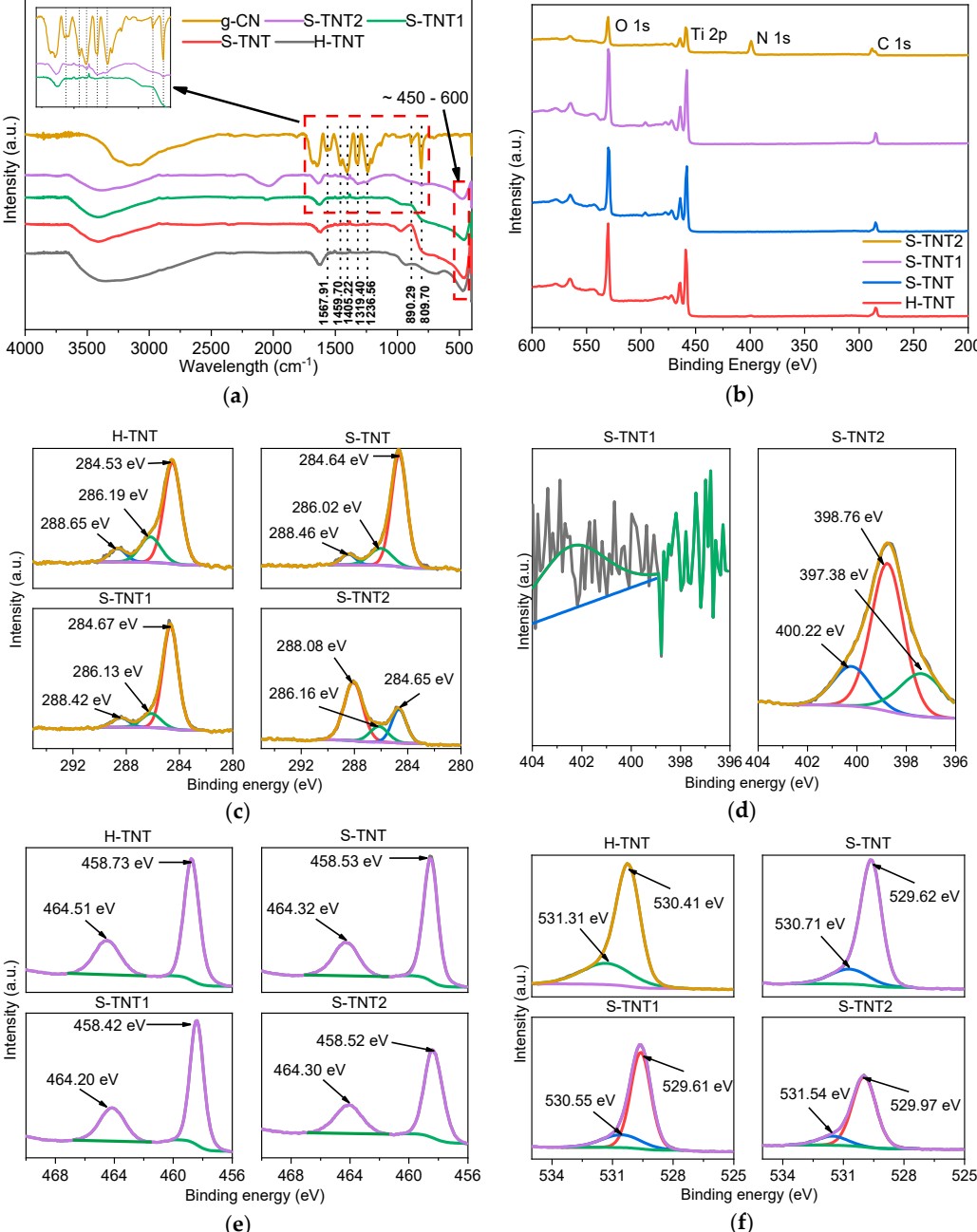

**Figure 4.** (**a**) Fourier transform infrared (FT-IR) spectra and (**b**) wide-angle survey scan of the as-prepared composites; high resolution survey scan of (**c**) C 1s, (**d**) N 1s, (**e**) Ti 2p, and (**f**) O 1s for H-TNT, S-TNT, S-TNT1, and S-TNT2.

To further evaluate the as-prepared single element and composite sample's chemical compositions, X-ray photoelectron spectroscopy (XPS) analysis was conducted. Figure 4b illustrated the broad-angle

scan, where Figure 4c,f presented the deconvoluted C 1s, N 1s, Ti 2p, and O 1s spectra of the as-prepared samples. The wide-angle survey scan shown in Figure 4b shows that the presence of Ti 2p, O 1s, and C 1s spectra was confirmed in the prepared samples. Additionally, only the composite S-TNT2 showed any wide scan peak for the binding energies of N 1s. For the prepared composites S-TNT1 and S-TNT2, the C 1s (Figure 4c) and N 1s (Figure 4d) spectra were investigated to confirm the formation of g-CN within the composite. From Figure 4d, characteristic N 1s peaks for g-CN were found in S-TNT2, but not in S-TNT1. A negligible amount of N atom was found in the composite, which may have been present in the doped condition. The FT-IR spectrum of S-TNT1 is also in coherence with these results. From the N 1s spectra of S-TNT2, three convoluted peaks were found around 397.38, 398.76, and 400.22 eV, which can be assigned to pyridinic N, pyrrolic N, and graphitic N, respectively [19,22]. Besides, from the C 1s scan (Figure 4c) of S-TNT2, a dominant peak for N-C=$N_2$ coordination was found around 288.08 eV [19]. Additionally, because of the incorporation of g-CN with S-TNT in S-TNT2, the Ti 2p and O 1s scans' peak positions showed a positive shift of ~0.1 eV. The absence of no visible peak for C-Ti confirmed that there were no chemical reactions between S-TNT and g-CN, and a rather successful heterojunction was prepared.

By investigating the Ti 2p narrow scan (Figure 4e) of the prepared H-TNT, sharp peaks were found around 458.73 eV and 464. 51 eV, which can be ascribed to the characteristic peaks of Ti $2p_{3/2}$ and Ti $2p_{1/2}$ [22]. Hence, it is safe to presume that the Ti in H-TNT is present in the $Ti^{4+}$ form. Besides, from the O 1s scan of H-TNT (Figure 4f), the peaks around 530.41 eV and 531.31 eV could be assigned for $O^{2-}$ lattice and $OH^-$, respectively, which confirms the formation of $TiO_2$ in H-TNT [49]. Similar peaks were observed in Ti 2p and O 1s scan of S-TNT; however, a negative shift of ~0.2 eV was observed due to the reduction of impurities, i.e., extra H and C atoms. Hence, based on the FT-IR and XPS results, it can be presumed that successful heterojunction was prepared in S-TNT2 without compromising any of their chemical features. However, a lower melamine content in the precursor mix (H-TNT/melamine = 2:1) of S-TNT1 caused N/C doped S-TNT synthesis, which has modified the optical and photocatalytic attributes of the sample.

### 2.4. Optical Traits (Photoluminescence (PL) and UV/Vis Diffuse Reflectance Spectroscopy (DRS))

It is necessary to evaluate the compound's inherent optical traits to assess the in-depth mechanism of photoactivity of the photoactive semiconductor. Hence, diffuse reflectance spectroscopy (DRS) UV/vis analysis was conducted to elucidate the amendment in the light absorption range and estimate the optical bandgap of the composites (S-TNT1 and S-TNT2). The findings associated with DRS UV/vis are depicted in Figure 5a–c. The absorbance spectra of S-TNT2 presented in Figure 5a showed distinct enhancement in the visible light region; hence, compared with S-TNT and S-TNT1, light absorption under UV spectra was reduced for obvious reasons. The distinction is more evident in Figure 5b, where the sequence of light absorption wavelength edges was found as follows: g-CN (452.62 nm) > S-TNT2 (434.10 nm) > S-TNT1 (404.92 nm) > S-TNT (403.00 nm). Using the Kubelka–Munk function and through the process of extrapolation, the optical bandgaps of the as-prepared samples were estimated and are depicted in Figure 5c [26,50]. Bandgap narrowing was observed in the prepared S-TNT2 compared with S-TNT and S-TNT1. Besides, the CB and VB positions of the semiconductors were determined using the following equations:

$$E_{CB} = E_{VB} - E_g \tag{1}$$

$$E_{VB} = X - E_0 + 0.5\,E_g \tag{2}$$

where $E_{CB}$ and $E_{VB}$ are CB and VB potential, respectively; $X$ is the mean (geometric mean of the component atoms) electronegativity of the semiconductor; $E_0$ is the energy of free electrons against the normal hydrogen electrode (NHE); and $E_g$ is the energy bandgap of the semiconductor. Based on the relevant articles, for anatase $TiO_2$ and g-CN, the value of electronegativity can be ascribed as 5.81 and 4.64 eV, respectively, and $E_0$ can be taken as 4.5 eV [36]. Hence, the $E_{VB}$ of S-TNT and g-CN can be

estimated as +2.90 eV and +1.5 eV, respectively; moreover, the corresponding $E_{CB}$ will be −0.28 eV and −1.22 eV, respectively.

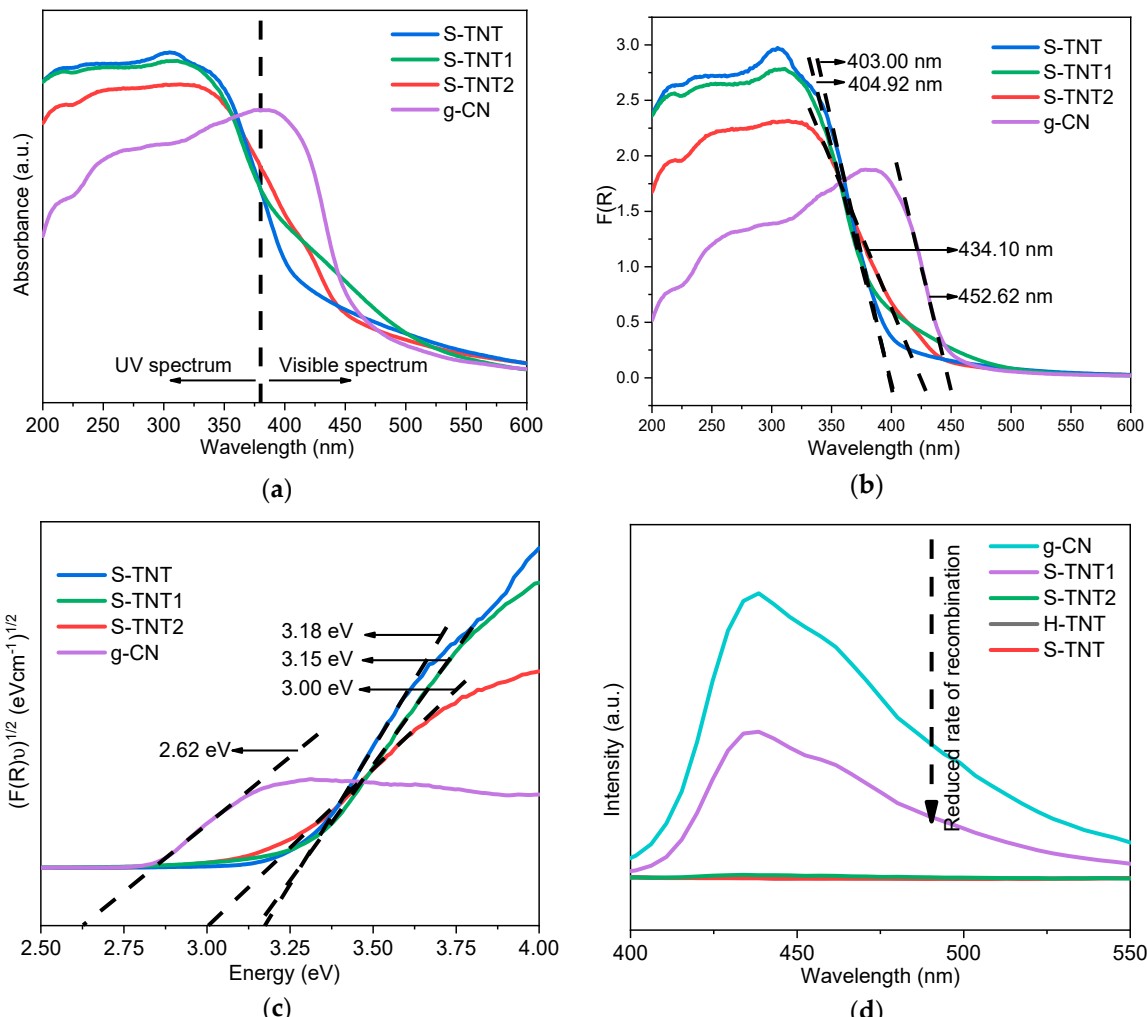

**Figure 5.** (**a**) UV/vis diffuse reflectance spectroscopy (DRS) (absorbance vs. wavelength), (**b**) UV/vis DRS (F(R) vs. wavelength), (**c**) converted Kubelka–Munk function vs. energy of absorbed light, and (**d**) photoluminescence (PL) spectra of the as-prepared S-TNT, S-TNT1, STNT2, and g-CN.

In addition, to assess the rate of recombination of the photogenerated $e^-/h^+$ pairs, PL analysis was conducted. Based on relevant studies, PL spectra have been found to be an effective strategy to assess the photoactivity of a semiconductor [22]. The estimated PL spectra of the as-prepared samples are presented in Figure 5d. The laboratory-made g-CN showed a semi asymmetric band around the maximum wavelength of ~439 nm, and similar peaks with reduced intensity were found for the prepared composites (S-TNT1 and S-TNT2). Based on the PL spectra, the recombination rate for S-TNT2 is minimal under visible light; hence, its photoactivity should be at a maximum. On the other hand, S-TiO$_2$ and H-TNT showed a straight line as PL spectra in the visible light region, which is evident as light absorption was minimal (Figure 5a).

## 2.5. Photoactivity

### 2.5.1. NO$_x$ Removal

The as-synthesized sample's photoactivity was evaluated by investigating the extent of atmospheric NO$_x$ removal under both UV and visible light irradiation in a continuous flow reactor. Moreover, the range

of $NO_x$ photooxidation was compared with commercially available NP400. Figure S2a and Figure 6a present the $NO_x$ removal patterns and overall $NO_x$ removal under 1 h of UV irradiation on the prepared samples, respectively. Compared with NP400, the prepared nanotubes showed a stable $NO_x$ removal pattern. Under UV irradiation, S-TNT2 was found to be most effective in removing atmospheric NO, around 32.44% NO removal with a nitrate selectivity of 48.19% (see Figure 6a). Compared with S-TiO$_2$, g-CN, and NP400, the extent of photooxidation using S-TNT under UV light was 6.4, 3.04, and 1.5 times higher, respectively. Similarly, under visible light, the prepared nanotubes showed a stable $NO_x$ removal pattern (Figure S2b), and S-TNT2 reported maximum removal. Under 1 h of visible light irradiation, S-TNT2 was found to remove 19.62% (Figure 6b) of NO, which was about 4.8, 3.7, 2.09, and 2.8 times higher than S-TiO$_2$, S-TNT, NP400, and g-CN, respectively. Moreover, under visible light, S-TNT2 showed superior nitrate selectivity of almost 60%.

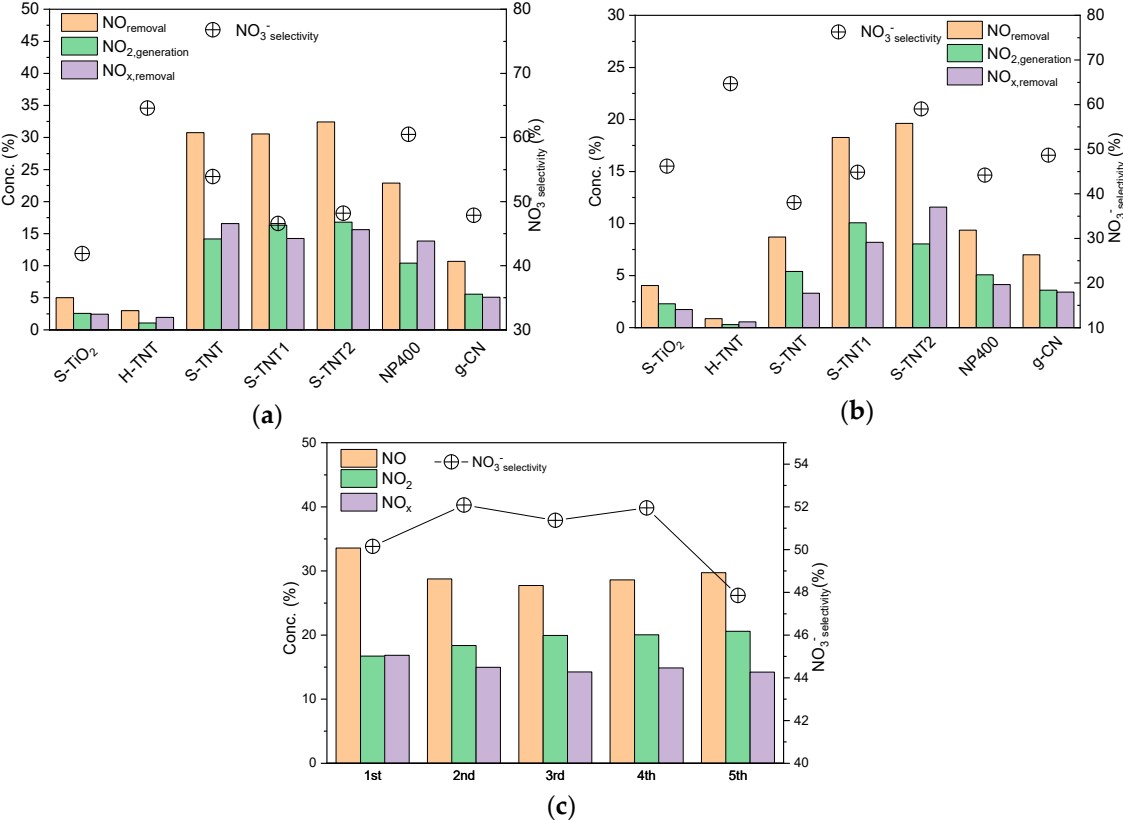

**Figure 6.** Eestimated $NO_{removal}$, $NO_{2,generation}$, $NO_{x,removal}$, and $NO_3^-$ selectivity profile of the as-synthesized samples along with NP400, under UV (**a**) and visible light (**b**); (**c**) recycling experiments for NO removal under UV irradiation over S-TNT2.

To further assess the stability of the prepared composite, recycle experiments up to five runs were carried out using UV irradiation on the as-prepared S-TNT2 (see Figure S2c). Each cycle was continued for 30 min with a 5 min stopping period that was considered between each run. After five cycles, the photoactivity showed negligible decay, demonstrating the high stability and reusability of the as-prepared composites. After 30 min of UV irradiation in the first run, total NO removal was estimated to be 33.58%, which was only reduced to 29.72% at the end of the fifth run (Figure 6c). Moreover, the mild decreasing pattern confirms the presence of NO oxidized by-products on the active photocatalytic sites. Interestingly, the nitrate selectivity was preserved in the first four runs (around 51%) and reduced slightly in the fifth run (47.85%).

### 2.5.2. Mechanism of NO$_x$ Removal

Based on the aforementioned advanced characterization of the as-prepared samples, three key reasons were found that contributed towards the enhanced photoactivity (under UV/visible irradiation) of the studied S-TNT/g-CN (S-TNT2) composite. These primary reasons are (a) enhancement of the effective surface area, (b) narrowed optical bandgap, and (c) reduced recombination rate of photogenerated e$^-$/h$^+$ pairs.

*(a) Enhancement of effective surface area:* The successful S-TNT/g-CN heterojunction was observed in the as-synthesized S-TNT2, which showed an effective surface area of 153.61 m$^2$/g. The pristine S-TiO$_2$ and S-TNT showed S$_{BET}$ values of 51.69 m$^2$/g and 88.55 m$^2$/g, respectively. Hence, during the photooxidation of NO$_x$ under either UV or visible light irradiance, S-TNT2 provided a significantly high amount of active photooxidation sites [50]. In the case of visible light irradiance, a high extent of NO$_x$ removal was evident, due to narrowed bandgap (Figure 5c), as well as a reduced charge recombination rate (Figure 5d) along with the increased active sites [27,32,37]. However, from the UV/vis DRS spectra, it was apparent that the UV regime's light absorption was reduced for S-TNT2 compared with S-TNT (Figure 5a). Irrespective of reduced UV light absorption, the prepared S-TNT2 showed enhancement in NO removal from 30.75% (S-TNT) to 32.44%, due to the dominant effect of effective surface area. Hence, visible light photo-activation of S-TNT/g-CN was possible without compromising the photoactivity under UV irradiance.

*(b) Narrowed optical bandgap:* Because of the incorporation of g-CN with S-TNT, the optical bandgap of S-TNT2 was reduced to 3.00 eV from 3.18 eV (S-TNT). As shown in Figure 5c, the optical bandgap of g-CN was reported as 2.62 eV; hence, the narrowing of bandgap in S-TNT2 may be attributed to the presence of g-CN in the composite [34,36]. Therefore, similar to g-CN, a redshift (Figure 5b) was observed in S-TNT2, leading to activation of photoactivity under visible light.

*(c) Reduced recombination rate of photogenerated e$^-$/h$^+$ pairs:* In comparison with g-CN, the as-prepared S-TNT2 showed reduced intensity in PL spectra, which indicates the extent of charge recombination is decreased [35]. Hence, the photoactivity of the prepared composites improved under both UV and visible light. Consequently, the trend of improvement in NO$_x$ removal under UV/visible light (Figure 6) and the increase in the intensity of the PL spectra (Figure 5d) showed a similar pattern (g-CN < S-TNT1 < S-TNT2).

It is generally accepted that photocatalytic NO oxidation mainly comprises the active species such as superoxide (O$_2$$^-$) and hydroxyl (OH) radicals produced from redox reactions involving photogenerated e$^-$/h$^+$ pairs and adsorbed O$_2$ and OH$^-$ groups on the irradiated photocatalyst [27,32]. For the photogenerated e$^-$ to produce O$_2$$^-$ effectively, the CB edge of the photocatalyst must be more negative than the redox potential of O$_2$/O$_2$$^-$ (0.33 eV vs. NHE). Alternatively, the standard redox potential for OH$^-$/OH is 1.99 eV against NHE; hence, the position of VB needed to be more positive for the h$^+$ to generate OH radicals. Based on the estimated CB and VB edge position of the prepared samples (S-TNT and g-CN), Figure 7 presents the proposed schematic for NO removal under UV/visible irradiance. The estimated optical bandgaps along with the CB and VB position of S-TNT and g-CN of this study suggested the as-synthesized composite to be of type II heterojunction photocatalyst [50].

Under UV irradiance, both S-TNT and g-CN in the composite (S-TNT2) were excited and generated photoinduced e$^-$ and h$^+$ in their respective CB and VB. As the CB of g-CN (−1.22 eV) is more negative compared with S-TNT (−0.28 eV), the photogenerated e$^-$ migrates towards the CB of S-TNT [27]. Later, e$^-$ reacts with surrounding O$_2$ and H$_2$O to generate ·O$_2$$^-$ and ·OH radicals. On the other hand, the photoinduced h$^+$ on the highly positive VB of S-TNT (+2.90 eV) migrates towards less positive VB of g-CN (+1.50 eV) and reacts with the adsorbed H$_2$O to generate OH radicals. Later, the generated ·O$_2$$^-$ and OH$^-$ radicals oxidize the NO into NO$_2$ or NO$_3$$^-$ [27,35,37]. Relevant works reported that, under both UV and visible light, the primary species for NO oxidation to NO$_3$$^-$ is the O$_2$$^-$ [27]. Hence, based on the reported nitrated selectivity of the prepared nanotubes (S-TNT = 53.90%, S-TNT1 = 46.59%, and S-TNT2 = 48.19%), it is safe to presume that both ·O$_2$$^-$ and OH radicals are primary

active species for NO oxidation under UV light. The following equations summarize the underlying reactions when UV light is irradiated upon S-TNT2.

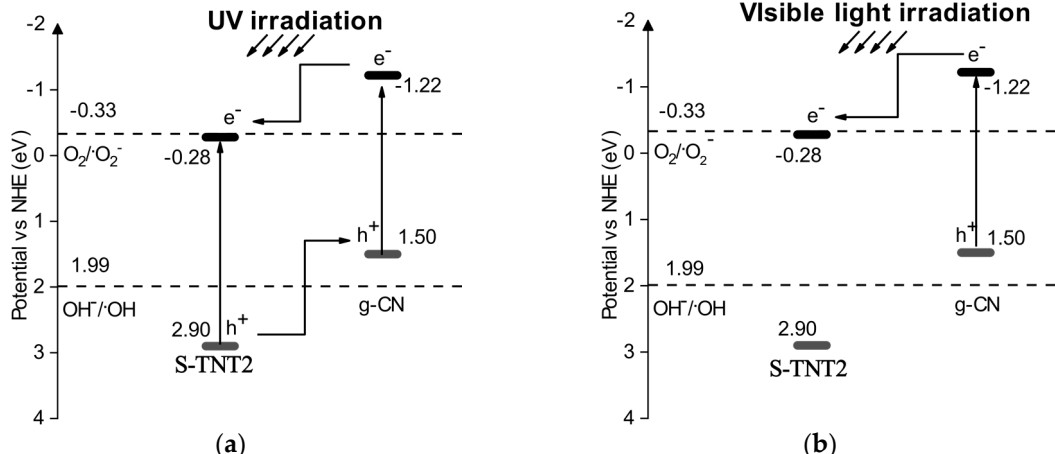

**Figure 7.** Proposed schematic of NO removal mechanism for the as-prepared S-TNT/g-CN composites, S-TNT2 under (**a**) UV irradiance and (**b**) visible light.

$$e^- + O_2 \ \rightarrow \ O_2^- \ (\text{Primary active species}) \tag{3}$$

$$e^- + O_2 + H_2O \ \rightarrow \ OH \ (\text{Primary active species}) \tag{4}$$

$$h^+ + OH^- \ \rightarrow \ OH \ (\text{Primary active species}) \tag{5}$$

$$OH + NO \ \rightarrow \ NO_2 \tag{6}$$

$$O_2^- + NO \ \rightarrow \ NO_3^- \tag{7}$$

Alternatively, under visible light irradiation on $TiO_2$/g-CN composites, only g-CN was excited, and the photogenerated $e^-$ can migrate to S-TNT and readily produce $O_2^-$, but the $h^+$ on the VB cannot directly oxidize $H_2O$/$OH^-$ to ·OH radicals [27]. Hence, $O_2^-$ should be the primary reason for NO oxidation under visible light. The estimated nitrated selectivity of the prepared nanotubes also supports the above hypotheses, and consecutive visible light activation of the composites showed an increasing trend (see Figure 6b). Therefore, Equations (3) to (7) summarize the underlying reactions of visible light induced photooxidation of NO. However, the discussion mentioned above confirmed that the $e^-$ and $h^+$ generated reactive species shown in Equations (4) and (5) are the secondary active species during photooxidation under visible light.

## 3. Materials and Methods

### 3.1. Materials

The precursors NP400 (anatase titania) and melamine ($C_3H_6N_6$, assay: 99%) in powder form were collected from Bentech Frontier Co. Ltd. (Gwangju, South Korea) and Sigma-Aldrich (Seoul, South Korea), respectively. For coagulation and flocculation, $TiCl_4$ (Assay: 99%, Sigma-Aldrich, Seoul, Korea) was procured. Additionally, for the hydrothermal treatment, sodium hydroxide (NaOH) (97%, w/w) and hydrochloric acid (HCl) (37%, v/v) were procured from the Sigma-Aldrich, Korea. Except for $TiCl_4$, all the chemical reagents were used without any further modification.

### 3.2. Preparation of S-TiO$_2$ and H-TNT

Dye wastewater was collected from a wastewater treatment plant situated in Daegu, South Korea, for Ti-based coagulation and flocculation. The wastewater's ambient physicochemical properties were

estimated as pH, 11.70; COD, 449 mg/L; TN, 72 mg/L; and TP, 3.2 mg/L. For coagulation, a 20 wt.% of TiCl$_4$ was prepared, and a conventional jar test was conducted with an initial rapid rpm of 100 for 1 min and subsequent slow rpm of 30 for 20 min. Later, the flocculated wastewater was collected and dried for 48 h in the laboratory oven at a controlled temperature of 100 °C. Then, the blackish dried flocculated sludges were powdered in a mortar and pestle. Finally, the milled powder was calcined at a constant temperature of 600 °C for 3 h in a box furnace at a heating ramp of 10 °C/min for the preparation of S-TiO$_2$.

For the nanotube, initially, 3 g of S-TiO$_2$ and 80 mL of NaOH (10 N) was placed in a glass beaker for 1 h using a conventional magnetic stirrer. Then, the uniform suspension was collected and placed in a teflon lined autoclave. Later, the laboratory oven temperature was prefixed at 180 °C, and the autoclaves were placed there for 48 h for hydrothermal treatment. After the completion of hydrothermal treatment, centrifugation at 3000 rpm was utilized to recover Na-titania nanotubes (Na-TNTs) from the residual suspension. The recovered solids were treated for ion exchange in a magnetic stirrer at pH 2 using 1 N HCl solution. After ion exchange, the sample was washed with ultra-pure water until the pH reached 7. Finally, H-titania nanotubes (H-TNTs) were prepared through the sample's overnight drying at 100 °C.

### 3.3. Preparation of S-TNT/g-CN Composite

To form a heterojunction, the dry mixture of H-TNT and melamine (Sigma-Aldrich, 99%) was calcined in a laboratory-scale box furnace at a temperature of 550 °C for 3 h. The rate of temperature rise in the furnace was maintained at 10 °C/min. Three samples were prepared by varying the melamine content from 0 to 100%. The salient features of the prepared samples are tabulated in Table 1.

**Table 1.** Nomenclatures and salient features of the preparation method for prepared composites. S-TNT, sludge generated titanium dioxide nanotube; H-TNT, H-titania nanotube.

| ID | Amount (g) | | Prepared Sample (g) | Mixing Ratio |
|---|---|---|---|---|
| | **H-TNT** | **Melamine** | | |
| S-TNT | 2.5 | - | 2.4 | - |
| S-TNT1 | 2.5 | 1.25 | 2.1 | 2:1 |
| S-TNT2 | 2.5 | 2.5 | 2 | 1:1 |

### 3.4. Characterization

XRD analysis was performed to characterize the crystalline structure of the prepared composites. MDI Jade 5.0 was utilized to generate the XRD patterns of the as-synthesized samples on D/MAX Ultima III, Rigaku, Japan. The Cu anode, along with Kα radiation, was employed in the diffractometer at a voltage of 40 kV and 40 mA current in the X-ray tube. By considering Bragg–Brentano geometry, and a scanning rate of 0.02/s, the XRD patterns were generated at a 2θ range of 2° to 90°. With a 0.3 mm receiving slit, the X-ray beam's divergent height and divergence were set as 10 mm and 2/3°, respectively.

SEM (S-4700, Hitachi, Minato-ku, Japan), running at 15 kV in a vacuum and working span of 7 mm, was employed to study powdered samples' morphology. An EDX detector (55VP SEM) operating at 15 kV was used to evaluate the elemental compositions. The TEM of the as-synthesized samples was generated on a field emission transmission electron microscope (FE-TEM, JEOL Ltd., JEM-2100F, Akishima, Japan). For TEM, the samples were dispersed in ethanol, and a droplet was placed on the carbon-coated Cu grid. Later, the sample droplet was evaporated under atmospheric conditions. Finally, TEM was generated by operating the FE-TEM at 200 kV. The samples' effective surface areas (S$_{BET}$) and pore size distributions (PSDs) were assessed by utilizing Belsorp mini II (BEL, Suminoe-ku, Japan), an automated surface area analyzer. For the physisorbed isotherm generation, N$_2$ gas was utilized, and the S$_{BET}$ and PSD (desorption isotherm at P/P$_0$ = 0.99) were estimated using the BET and BJH models, respectively.

As a means of delineating the g-CN and S-TiO$_2$ in the as-synthesized samples, Fourier transform infrared (FT-IR) spectra were generated on IRPrestige-21 (Shimadzu, Kyoto, Japan). Light absorption was measured in the mid-infrared (4000–400 cm$^{-1}$) range. For sample preparation, the KBr pellet preparation method was utilized. The dry mix of milled samples and KBr was pressed at 5000–6000 psi to prepare the pellet. Additionally, using dry samples, XPS spectra were generated on the Kratos Axis Ultra X-ray Photoelectron Spectroscopy system (MultiLab 2000, VG, UK) to determine the varying chemical bonds present in the samples. For a qualitative assessment of the charge separation rate during photocatalysis, a fully automated spectrofluorometer system (Dual FL, Horiba, Piscataway, NJ, USA) was utilized to generate the photoluminescence (PL) spectra of the prepared samples. The DRS of the as-synthesized samples was carried out at KBSI Daegu center, South Korea, on a UV/vis/NIR spectrometer (Lambda 950, Perkin Elmer, KBSI daegu center, South Korea). The reflectance spectra were generated at a spectral range of 400 to 800 nm at a scanning rate of 600 nm/min. Finally, the optical band gap of each sample was determined using the Kubelka–Munk function (F(R)).

*3.5. Photocatalytic Activity*

A laboratory-scale photocatalytic reactor was utilized to assess the extent of photooxidation of the prepared samples through photooxidation of NO$_x$ under both UV and visible light. For UV light-induced NO$_x$ removal, ISO 22197-1 [40] was followed, where ISO 17198-1 [41] was followed for visible light irradiated NO$_x$ removal. For UV irradiation, according to ISO 4892-3, so-called black light (UV-A, Sanyo-Denki, Nagano, Japan) with a wavelength range of 300 nm to 400 nm (maximum 351 nm) was utilized. Additionally, cool fluorescent light with a UV cut-off filter for wavelengths below 400 nm was employed for the visible light, which adheres to ISO 14605. The experimental setup has been previously documented [51,52]. The schematic of the utilized system to evaluate the photocatalytic performance of the as-prepared samples is illustrated in Figure 8. Following the ISO protocols, the utilized photoreactor can hold the test sample within a 50 mm wide trough having a surface parallel to an optical window for photoirradiation. The sample piece was separated from the window by a 5 ± 0.5 mm thick air layer. The gap was carefully controlled using height adjusting plates, as shown in Figure 8. Then, 1 g of sample was mildly pressed in a rectangular sample holder with a surface area of 50 cm$^2$. For the removal of organic matters from the as-prepared samples, the sample holders with samples were pretreated under UV irradiance for 5 h at a light intensity of 10 W/m$^2$. Finally, the sample holder (with sample) was placed in the reactor attached with an NO$_x$ analyzer (CM2041, Casella, London, UK) to record the variation in NO, NO$_2$, and NO$_x$ concentrations profile during light (UV/visible) irradiation. The experimental parameters were locked according to the aforementioned ISO protocols. Throughout the experiments, the flow of NO at the inlet of the reactor was maintained as 1 ppmv with an airflow of 3 L/min. Besides, the moisture content and the internal temperature of the reactor were maintained as 50% and 25 °C, respectively. Each experiment was carried out for 100 min, 20 min of gas adsorption followed by 1 h of light irradiation, and 20 min of purging. During UV irradiated NO$_x$ oxidation, light intensity was maintained as 10 W/m$^2$ ± 0.5 W/m$^2$, and during visible light irradiance, it was 6000 lx ± 300 lx according to the followed ISO protocols. The average values of the three experiments were reported. The rate of NO removal, NO$_x$ removal, NO$_2$ generation, and nitrate (NO$_3^-$) selectivity was confirmed according to the following equations:

$$NO_{removal} = \frac{NO_{in} - NO_{out}}{NO_{in}} \tag{8}$$

$$NO_{2,generation} = \frac{NO_{2,out} - NO_{2,in}}{NO_{in}} \tag{9}$$

$$NO_{x,removal} = \frac{NO_{x,in} - NO_{x,out}}{NO_{in}} \tag{10}$$

$$NO_{3}^{-}{}_{\text{selectivity}} = \frac{NO_{x,\text{removal}}}{NO_{\text{removal}}} \times 100\% \qquad (11)$$

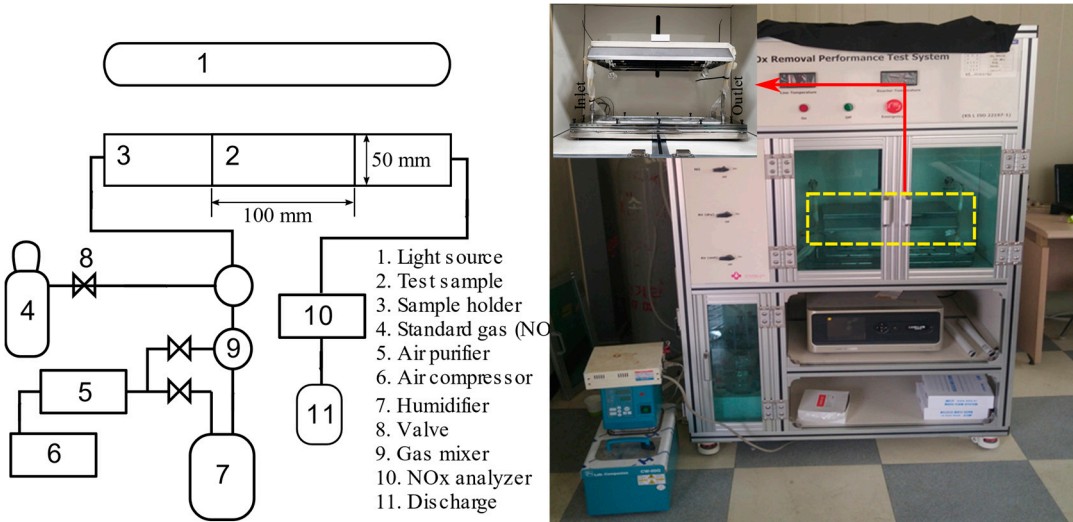

1. Light source
2. Test sample
3. Sample holder
4. Standard gas (NO
5. Air purifier
6. Air compressor
7. Humidifier
8. Valve
9. Gas mixer
10. NOx analyzer
11. Discharge

**Figure 8.** The schematic of the experimental setup (**left**) along with the photo of the experimental setup (**right**) for photocatalytic NO removal under UV and visible light irradiance.

## 4. Conclusions

The study reported the successful application of an NaOH-based hydrothermal treatment for the preparation of sludge generated $TiO_2$/g-CN nanotubes. Through advanced characterization and UV/visible light induced photooxidation of NOx in a continuous flow reactor, the following hypotheses were deduced:

- NaOH-based modified hydrothermal treatment of $TiO_2$ generated from sludges of dye wastewater can produce H-titanate after successive acid wash and lead to the preparation of pure anatase $TiO_2$ nanotubes through calcination.
- Facile calcination of sludge generated H-titanate, and an equal part of melamine mix at a temperature of 550 °C for 3 h can successfully produce anatase $TiO_2$/g-CN nanotubes with severely increased effective surface area compared with the pristine nanotubes prepared by only calcining H-titanate.
- Owing to bandgap narrowing, the prepared $TiO_2$/g-CN nanotubes showed a redshift in light absorption (434.10 nm). Hence, UV light absorption was reduced compared with pristine $TiO_2$ nanotubes, but the several-fold increase in effective surface area counterbalanced that drawback.

Under visible light irradiation of 1 h, the as-synthesized $TiO_2$/g-CN nanotubes showed almost 20% of NO removal with an enhanced nitrate selectivity of 59.02%, which is a significant improvement compared with pristine sludge generated TiO2 nanotubes.

**Supplementary Materials:** The following are available online at http://www.mdpi.com/2073-4344/10/11/1350/s1, Figure S1: EDX spectra of (a) S-TiO$_2$, (b) H-TNT, (c) S-TNT, (d) S-TNT1, and (e) S-TNT2., Figure S2: Concentration patterns of NO, NO$_2$, and NO$_x$ in the continuous flow reactor of the as-synthesized samples along with NP400, under UV (a) and visible light (b); (c) recycling experiments for NO removal under UV irradiation over S-TNT2; (d) NO$_x$ concentration profile under light irradiation (UV/visible) without the presence of any photocatalysts.

**Author Contributions:** Conceptualization and methodology, S.M.H., H.P., and J.S.M.; formal analysis, investigation, and data curation, S.M.H., H.-J.K., and I.R.; writing—original draft preparation, S.M.H.; writing—review and editing, L.T., Y.-S.J., and H.K.S.; supervision, J.-H.K. and H.K.S. All authors have read and agreed to the published version of the manuscript.

**Funding:** This research was supported by a grant (18SCIP-B145909-01) from Smart Civil Infrastructure Research Program funded by Ministry of Land, Infrastructure, and Transport of the Korean government and the National Research Foundation of Korea (NRF) grant funded by the Korean government (MSIT) (No. 2019R1C1C1007745).

**Conflicts of Interest:** The authors declare that they have no known competing financial interests or personal relationships that could have appeared to influence the work reported in this paper.

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
