# Peer review of "Modified Hydrothermal Route for Synthesis of Photoactive Anatase TiO2/g-CN Nanotubes from Sludge Generated TiO2"

_catalysts, doi:10.3390/catal10111350_

Round 1
Reviewer 1 Report
The article is very interesting, well written and the results are argued by different methods of characterization. However, attention must be paid to the reference lists cited. Below are some comments to improve the manuscript:
1-Page 2:
-Line 69-70: in the article of Kasuga, surface area is 400 m²/g and nanotube diameter is 8 nm
-Line 71-72: you write “As TNTs have a wide variety of applications, they were expected to have a more beneficial impact for photocatalytic applications than other forms of nanotubes such as nanoparticles, colloidal,..”, nanoparticles and colloidal are not nanotubes. You can write …other forms of particles….
-Line 74: titanate rather than titanite.
-Line 77-80: You quote the reference Okour [26], I think there is confusion. The article mentioned does not deal with mesoporous structure.
2-Page 3-Line 131-146: Figure 1a shows a small peak to the right of the peak (101) at 25.50 degrees for the S-TiO2 sample. This peak apparently does not exist for the other sample. Can you explain why?
3-Could you argue the choice of the NP400 comparison sample?
4-Page 14-Line 380: can’t rather than cant.
5-Page 14-Line 386-389: the equations (8) to (11) are the same than (4) to (7). You can remove (8) to (11) and refer to eq (4) to (7).
6-Page 14-Line 406-408: Beware of units, leaving a space between the numerical value and the unit symbol.
7-Page 16:
-Line 458: could you give a scheme or photo of the photocatalytic reactor? What is the volume of reactor?
-Line 461-462: Could you still give the wavelengths of the UV and visible lamps that you used?
-Line 464-465: Why do you pretreat sample holders with samples under UV irradiance for 5 hours?
-Have you done tests with the reactor without photocatalyst to see the effect of radiation on the NO directly?
-I don't know why you talk about NOx removal while you inject NO. Did you do the same thing by injecting NO2?
-The tests were carried out in a dynamic regime with a large airflow (3 L/min). The contact time between the NO and the material may be short. As a result, removal activity may be underestimated.
Reviewer 2 Report
Manuscript catalysts-977125 deals Modified hydrothermal route for synthesis of photoactive anatase TiO2/g-CN nanotubes from sludge generated TiO2. The work reported a detailed characterization of the prepared materials, however a considerable improvement can be taken into account in order to promote the quality of the work.
- The novelty carried out with this work was not reported and would be emphasized.
- In the abstract section, a comment about photocatalytic activity could be mentioned.
- In Table 1 the last columns were not necessary since the T, ramp rate, and duration are the same for all lines. This information could be included in the text form during the manuscript.
- In the line 442 appeared SBTE instead of SBET.
- EDX analysis allowed to perform a qualitative analysis and not a quantitative analysis as reported in the manuscript. EDX analysis were carried out in some part of the material and if it is not very homogenous the composition is very variable and therefore a quantification cannot be withdrawn.
- The symbol of NO- selectivity is very confused. The authors could change for a simple symbol, as a circle or a square.
- In line 374, 386, and 387 the number of equations was not formatted.
- Figure 6 is very confused. In order to simplify the information reported here, some figures can be removed as the concentration in ppm during time presented in a), c) and part of e).
- Figure 7 is also very confused. A new organization of the data reported could be an option.
Round 2
Reviewer 2 Report
The authors took into account the comments, therefore the paper can be published in the present form.